# The Seasonality of Respiratory Viruses in a Hong Kong Hospital, 2014–2023

**DOI:** 10.3390/v15091820

**Published:** 2023-08-26

**Authors:** Wai-Sing Chan, Siu-Kei Yau, Man-Yan To, Sau-Man Leung, Kan-Pui Wong, Ka-Chun Lai, Ching-Yan Wong, Chin-Pang Leung, Chun-Hang Au, Thomas Shek-Kong Wan, Edmond Shiu-Kwan Ma, Bone Siu-Fai Tang

**Affiliations:** Department of Pathology, Hong Kong Sanatorium & Hospital, Hong Kong SAR, China; waising.chan@connect.polyu.hk (W.-S.C.); yausiukei179@gmail.com (S.-K.Y.); renevanilla@yahoo.com.hk (M.-Y.T.); rositawai@gmail.com (S.-M.L.); pui_90123@yahoo.com.hk (K.-P.W.); chris_4253@yahoo.com.hk (K.-C.L.); joanne_ice@hotmail.com (C.-Y.W.); alex.cp.leung@hksh.com (C.-P.L.); tommy.ch.au@hksh.com (C.-H.A.); thomas.sk.wan@hksh.com (T.S.-K.W.); eskma@hksh.com (E.S.-K.M.)

**Keywords:** COVID-19, influenza, PCR, prevalence, respiratory virus, SARS-CoV-2, seasonality

## Abstract

We reviewed the multiplex PCR results of 20,127 respiratory specimens tested in a hospital setting from January 2014 to April 2023. The seasonal oscillation patterns of 17 respiratory viruses were studied. Compared with 2014–2019, a prominent drop in PCR positivity (from 64.46–69.21% to 17.29–29.89%, *p* < 0.001) and virus diversity was observed during the COVID-19 pandemic, with predominance of rhinovirus/enterovirus, sporadic spikes of parainfluenza viruses 3 and 4, respiratory syncytial virus and SARS-CoV-2, and rare detection of influenza viruses, metapneumovirus, adenovirus and coronaviruses. The suppressed viruses appeared to regain activity from the fourth quarter of 2022 when pandemic interventions had been gradually relaxed in Hong Kong. With the co-circulation of SARS-CoV-2 and seasonal respiratory viruses, surveillance of their activity and an in-depth understanding of the clinical outcomes will provide valuable insights for improved public health measures and reducing disease burden.

## 1. Introduction

The coronavirus disease 2019 (COVID-19) pandemic has affected not only the health of hundreds of millions globally [1] but also brought along sociological impacts from regional to individual levels [2,3,4]. In Hong Kong, having experienced outbreaks of severe acute respiratory syndrome (SARS) in 2003 [5] and human swine influenza pandemic in 2009 [6], citizens and government have been vigilant since the first cases of COVID-19 were recognized [7]. Countermeasures have been implemented by policy makers at the early stage and in response to the dynamic situation of the pandemic, such as quarantine of infected individuals and their close contacts; social distancing; restriction of indoor, outdoor, and school activities; arrival restrictions; mandatory mask wearing; and, more recently, lifting of these countermeasures [8]. The timeline of a few of these countermeasures is given in Figure 1. There were reports from a number of regions on the seasonality of respiratory viruses before and during the COVID-19 pandemic, revealing suppression of non-SARS-CoV-2, clinically important respiratory viruses during the period of unusual, global-scale pandemic intervention and their resurgence after ‘returning to normal life’ [9,10,11,12]. This knowledge provides valuable insights for better preparedness against future outbreaks and pandemics, which is especially important to healthcare providers for situation assessment and wise allocation of manpower and resources to cope with surges of clinical needs. In light of this, we reviewed the multiplex polymerase chain reaction (PCR) results of the past 10 years to study the seasonal oscillation patterns of respiratory viruses before and during the COVID-19 pandemic.

## 2. Materials and Methods

### 2.1. Clinical Specimens

A total of 20,127 respiratory specimens were collected from 18 January 2014 to 30 April 2023, with 590 from lower respiratory tract (LRT, including sputa and endotracheal/tracheal/bronchial aspirates), 16,624 from upper respiratory tract (URT, including swabs or aspirates of nasal cavity/nasopharynx/throat and posterior oropharyngeal saliva), and 2913 combined URT (combined nasal/nasopharyngeal and throat swabs) specimens. All specimens were sent to Department of Pathology, Hong Kong Sanatorium & Hospital, where multiplex PCR assays were performed for detection of respiratory viruses, as described below.

### 2.2. Multiplex PCR Assays for Respiratory Viruses

Two commercial multiplex PCR assays were used with reference to manufacturers’ recommendations, namely Luminex NxTAG Respiratory Pathogen Panel (NxTAG RPP, Luminex Molecular Diagnostics, Toronto, ON, Canada) and BioFire FilmArray Respiratory Panel (FA RP, BioFire Diagnostics, Salt Lake City, UT, USA). NxTAG RPP was used from January 2014 to June 2017, followed by FA RP hitherto. Viral targets included influenza A (flu A, subtypes H1, H1N1-pdm09, and H3) and B viruses (flu B), parainfluenza viruses 1–4 (PIV1–4), respiratory syncytial virus (RSV), metapneumovirus (MPV), adenovirus (ADV), rhinovirus/enterovirus (RV/EV), and coronaviruses (CoV) 229E, OC43, NL63, and HKU1. In addition, we also considered the data of Middle East respiratory syndrome coronavirus (MERS-CoV) and severe acute respiratory syndrome coronavirus 2 (SARS-CoV-2) from January 2021 to April 2023, which are additional targets in newer versions of FA RP.

### 2.3. Data Retrieval and Analysis

Data were extracted from the laboratory reporting system with specimen type confined to respiratory specimens. Specimens with indeterminate results were excluded. Personal identifiers of patients were removed prior to retrospective analysis of the data. Background demographic data on patient age, gender, and type (in- and outpatients) were considered. Clinical data on the PCR detection frequency of individual viruses per month and specimen type (LRT, URT, and combined URT specimens), co-detection frequencies and combinations, and the seasonal oscillation patterns were analyzed.

## 3. Results

### 3.1. Background Demographics

Demographic features of the study population per year are shown in Table 1. The patients were divided into six age ranges with reference to the US National Institutes of Health (NIH) [13]. Children (1 to 12 years) have been the leading age group, comprising 37.15–76.88% of the study population. Infants (1 month to <1 year, 5.97–12.35%), adults (18 to 64 years, 4.58–25.63%), and older adults (65 years or older, 4.21–25.70%) have comprised another significant portion. The proportion of males (50.50–54.12%) was slightly higher than females (45.88–49.50%). The proportion of inpatients has been decreasing from 80.96% to 44.60%, whereas that of outpatients has been rising from 19.04% to 55.40% over the 10 years.

### 3.2. Trends in the Number of Testing and PCR Positivity

From Figure 1, the number of specimens tested had been rising from 2014 to 2019. The monthly average rose from 136.58 in 2014 to 267.75 in 2019, and in general, the peaks of testing were in the first quarters (Q1) except for 2017, which was in the middle of the year. This rising trend ceased during the pandemic, with the monthly average dropping to 128.08 in 2020 and further to 82.83 in 2022. Except for Q1 of 2020, the peaks of testing were observed in Q4. The number of tests rebounded to 365 in March and 378 in April 2023.

The monthly average of PCR positivity was 64.46–69.21% in 2014–2019, dropped to 17.29–29.89% in 2020–2022, and rebounded to 72.88% and 67.72% in March and April 2023, respectively. In general, PCR positivity peaked in Q1 and Q4. Overall, the observed means in the number of testing and PCR positivity were significantly lower in 2020–2022 compared with 2014–2019 and 2023 (*p* < 0.001).

### 3.3. Co-Detection of Viruses

A single virus was detected in 74.02–87.28% (monthly average) of PCR-positive specimens in 2014–2019, 89.39–94.28% in 2020–2022, and 83.32% in 2023. For two viruses, the monthly average was 11.85–20.36% in 2014–2019, 5.57–8.82% in 2020–2022, and 15.13% in 2023. For 3–6 viruses, the monthly average was 0.87–5.63% in 2014–2019, 0.06–1.79% in 2020–2022, and 1.56% in 2023. Table 2 shows the frequency of co-detection for different combinations of respiratory viruses. The top three combinations were RV/EV-PIV4 (672), RV/EV-ADV (413), and RV/EV-RSV (396). Flu A virus subtype H1 and MERS-CoV were not detected throughout the study.

### 3.4. Specimen Type and Frequency of Detection

Table 3 shows the frequency of PCR detection in LRT, URT, and combined URT specimens. For all viruses except SARS-CoV-2, the frequency of detection in URT specimens was higher than that of LRT or combined URT specimens. For SARS-CoV-2, LRT specimens revealed the highest frequency.

### 3.5. Seasonality of Respiratory Viruses

The relative proportion of respiratory viruses is revealed in Figure 1. The seasonal patterns were similar throughout 2014 to 2019. Q1 was predominated by flu A and B viruses and RV/EV in Q2–Q4, except for 2017, when flu viruses peaked in July. This pattern was altered during the pandemic, with rare detection of flu viruses, MPV, ADV, and CoVs. RV/EV was predominant with sporadic spikes of PIV3 and 4, RSV, and SARS-CoV-2. The suppressed viruses appeared to regain activity from Q4 of 2022. The virus-specific seasonality is shown in Figure 2 and will be further discussed below.

#### 3.5.1. Influenza A and B Viruses

The 10-year mean of percentage total was 10.33%. Flu viruses displayed a pattern typical of ‘winter viruses’ in the pre-COVID-19 period, with major peaks observed in Q1 and minor peaks in June–September. The major peaks were, on average, 28.50% (25.32–43.14%) higher than the 10-year mean. The peaks were roughly type-specific. The frequency of detection was very low from February 2020 to December 2022 (0–3.55%) and rebounded to 19.05% in April 2023, with subtype H1pdm09 being the majority. Subtype H1 was not detected throughout the study.

#### 3.5.2. Parainfluenza Viruses 1–4

The 10-year mean was 8.70%. In 2014–2019, peaks were primarily observed in Q2 and Q4. Unlike flu viruses, the peaks showed a smaller difference from the 10-year mean (9.94% on average (7.74–12.13%)). No distinctive type-specific pattern was noted. PIV2 appeared to be less prevalent than the other three types. Frequency of detection was very low from February 2020 to September 2021 (0–4.90%), followed by three spikes predominated by PIV3 (44.86% in November 2021; 18.59% in February 2023) and PIV4 (10.77% in July 2022).

#### 3.5.3. Respiratory Syncytial Virus

The 10-year mean was 8.12%. In 2014–2016, spikes of RSV were observed every 4–6 months from April 2014 to November 2016 (12.44–19.44%). In 2017–2019, spikes were observed in September (12.78–28.44%). The frequency of detection was very low from April 2020 to October 2021 (0–7.32%) and March to October 2022 (0–1.92%), peaking at 25.16% and 24.37% in December 2021 and 2022.

#### 3.5.4. Metapneumovirus

The 10-year mean was 3.71%. From 2014 to 2019, detection of MPV peaked in March–June (10.14–18.06%). During the pandemic, MPV was not detected for 29 months (March 2020–July 2022). The percentage total rose gradually from August 2022 to a peak of 7.34% in November 2022.

#### 3.5.5. Adenovirus

The 10-year mean was 6.23%. The seasonality of ADV was not distinctive. In 2014–2019, peaks were observed from June to December (9.62–20.63%). The frequency of detection dropped below the 10-year mean from 2020 (0–5.80%).

#### 3.5.6. Rhinovirus/Enterovirus

The 10-year mean was 23.78%. Detection of RV/EV peaked in March–June and September–December (25.00–50.61%). A very low prevalence of RV/EV was observed from February to October 2020 and from February to May 2022 (0–6.45%).

#### 3.5.7. Coronaviruses and SARS-CoV-2

The 10-year mean was 4.35%. In 2014–2019, detection of CoVs peaked in January–March and August–December (6.90–13.40%). Spikes were type-specific, with NL63 and OC43 predominating in alternate years. The detection frequency of CoVs by multiplex PCR, including SARS-CoV-2, was very low from April 2020 to January 2022 (0–1.59%). A sharp peak of SARS-CoV-2 was observed in February 2022 (24.00%), and its predominance continued until February 2023, followed by the rebound of OC43 in March and April.

## 4. Discussion

The climate of Hong Kong is subtropical. Mean temperature is generally the highest in June–August (28–30 °C) and the lowest in December–February (16–18 °C) [14]. From our data, the peaks of testing and PCR positivity were observed in Q1 and Q4, which coincided with the phenomenon of winter epidemics in temperate regions [15]. Since early 2020, a number of countermeasures have been employed in Hong Kong to combat the spread of COVID-19. During the pandemic, the number of multiplex PCR tests dropped by at least twofold. PCR positivity and virus diversity also dropped in this period, primarily due to the rare detection of flu and other respiratory viruses. RV/EV was least suppressed, which might be due to the ineffectiveness of alcohol-based hand sanitizers and a weakened inhibitory effect of face masks against rhinovirus [16,17]. Up until March 2023, the majority of pandemic restrictions were lifted in Hong Kong, with the World Health Organization declaring an end to COVID-19 as a global health emergency in May 2023 [18]. A rebound of respiratory virus activity was observed in March and April 2023. Overall, our observation was similar to the change in seasonality in other geographical regions [9,10,11,12].

Indeed, this global-scale pandemic has enlightened us in several aspects. From the public health perspective, it provided evidence, albeit indirectly, that the collection of countermeasures might be effective against most of the clinically significant respiratory viruses, especially for the genetically variable flu A virus. This valuable experience will be useful for strategic planning against future outbreaks. From a healthcare provider’s perspective, the data on temporal changes in demographic features and local seasonality of respiratory viruses provide guidance on the optimal allocation of manpower and resources. For example, the demand for multiplex PCR testing had shifted towards SARS-CoV-2-specific PCR during the pandemic, with more than 5000 specimens on average per month. This drastic increase has driven us to respond quickly to manpower reallocation, compete for the scarce laboratory reagents, and adapt to the continuously updating anti-pandemic policies. The takeaway is a practical (or numerical) idea on the heightened manpower and resource requirement during a pandemic. From the scientific perspective, we could appreciate some interesting features about the seasonality of respiratory viruses. For instance, previous studies suggested a negative interaction between rhinovirus and influenza virus [19,20], which was also observed from our pre-COVID-19 data. Interestingly, during the period of very low/zero prevalence of the influenza virus, a lower RV/EV positivity was also noted in Q1 of 2021, 2022, and 2023. In addition, co-detection of RV/EV and flu viruses was the fourth most frequent. More research effort is warranted to decode the complex mechanism behind the seasonal oscillation pattern. From the clinical perspective, further study on the clinical outcome of co-detection cases may help identify the combinations with poor prognosis. For instance, Agathis and colleagues reported that among children less than 5 years old hospitalized with SARS-CoV-2 infection, co-detection with RSV or RV/EV was significantly associated with severe illness [21]. Amarin and colleagues observed that for 0- to 4-year-old children, co-detection of RSV with PIV or RV/EV was associated with lower odds of hospitalization and higher odds of intensive care unit admission for RSV-SARS-CoV-2 combination [22]. A better understanding in this area may facilitate evidence-based prediction of clinical outcomes and improved patient management.

Our study had several limitations. Technically, the multiplex PCR assays could not distinguish between RV and EV. We could not determine the viability of detected viruses nor identify co-infection events from co-detection data. Regarding specimen collection, as the majority of combined URT specimens were collected during the period of very low prevalence of non-SARS-CoV-2 respiratory viruses (2906 specimens collected after January 2020, 99.76%), the detection frequency of such might be underestimated. To a certain extent, the drastic shift in the demand towards SARS-CoV-2-specific PCR might introduce sampling bias, which might explain the very low prevalence of SARS-CoV-2 in 2021. Another source of sampling bias might originate from ‘duplicate’ specimens of the same patient collected during the same phase of PCR positivity. On the other hand, as the current study was based on observation of laboratory data, we could not determine the exact social factors contributing to the decreased number of specimens and PCR positivity during the pandemic. In addition, the data presented here were generated from a single hospital setting. A more extensive geographical coverage is warranted for a more representative display of the seasonality of respiratory viruses in Hong Kong.

## 5. Conclusions

At the time of writing, our life has returned to normal. The precious experience acquired in the past three years will guide us on how to cope with future pandemics wisely and promptly. With the co-circulation of SARS-CoV-2 and seasonal respiratory viruses, surveillance of their activity and in-depth understanding of the clinical outcomes will provide valuable insights for improved public health measures and reducing disease burden.

## Figures and Tables

**Figure 1 viruses-15-01820-f001:**
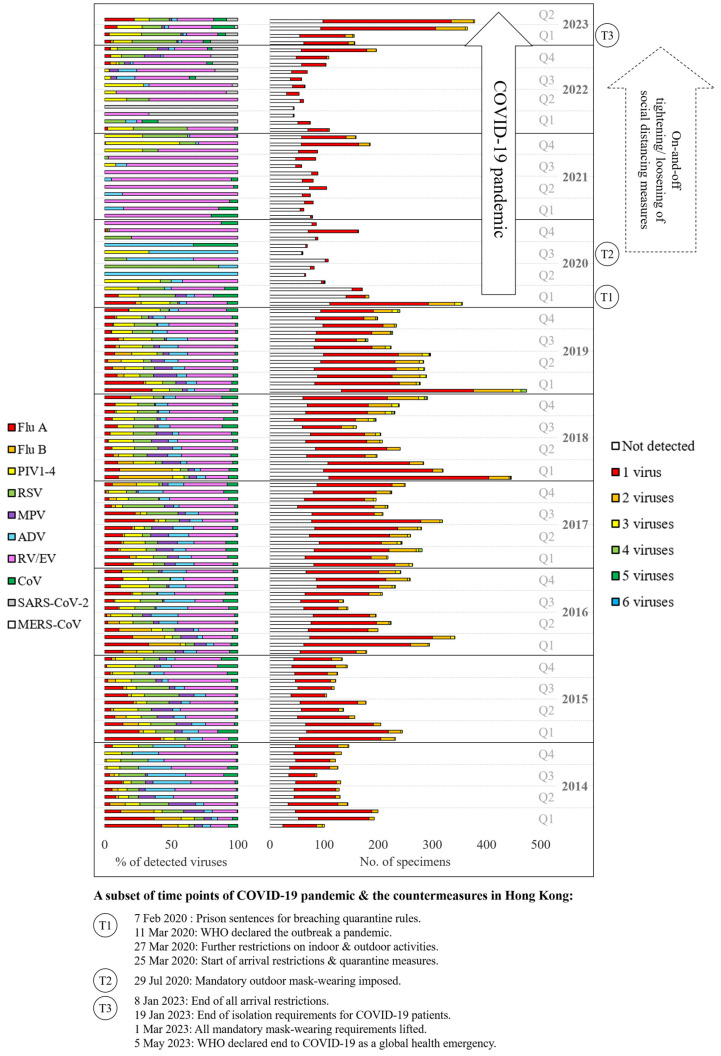
Seasonality of respiratory viruses from 2014 to 2023. The 100% stacked bar chart on the left shows the relative proportion of individual viruses (or virus groups). The composite bar chart on the right reveals monthly number of respiratory specimens tested and PCR positivity, classified by single-virus and co-detection. A subset of time points of the COVID-19 pandemic and the countermeasures in Hong Kong are marked by schematic diagrams. ADV, adenovirus; CoV, coronaviruses 229E, HKU1, NL63, and OC43; Flu A, influenza A virus subtypes H1, H1pdm09, and H3; MPV, metapneumovirus; MERS-CoV, Middle East respiratory syndrome coronavirus; PIV1–4, parainfluenza viruses 1–4; Q, quarter; RSV, respiratory syncytial virus; RV/EV, rhinovirus/enterovirus; SARS-CoV-2, severe acute respiratory syndrome coronavirus 2.

**Figure 2 viruses-15-01820-f002:**
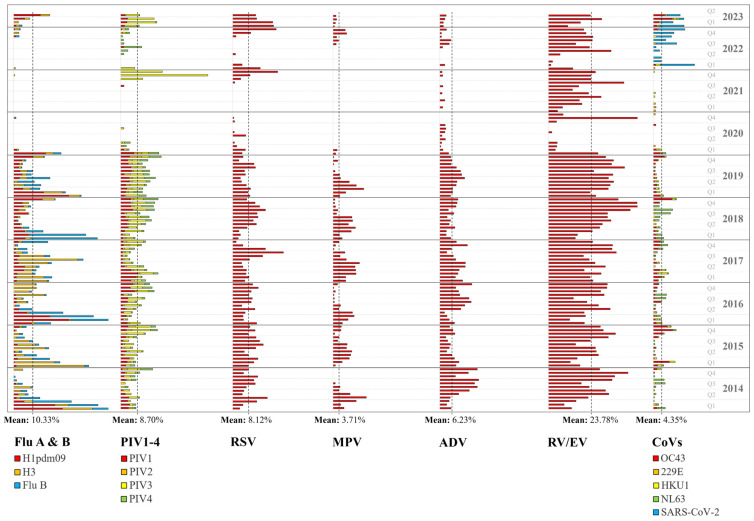
Monthly prevalence of individual viruses in percentage of specimens. The 10-year mean of each virus (group) is marked by dotted lines.

**Table 1 viruses-15-01820-t001:** Demographic features of the study population.

		* Year									
		2014	2015	2016	2017	2018	2019	2020	2021	2022	2023
**# NIH style** **age groups**	**Neonates or newborns** **(birth to <1 month)**	27 (1.65%)	32 (1.68%)	36 (1.35%)	20 (0.67%)	22 (0.73%)	22 (0.68%)	7 (0.46%)	9 (0.78%)	13 (1.31%)	4 (0.38%)
	**Infants** **(1 month to <1 year)**	184 (11.23%)	235 (12.35%)	224 (8.43%)	298 (10.05%)	311 (10.30%)	205 (6.38%)	129 (8.39%)	113 (9.84%)	89 (8.95%)	63 (5.97%)
	**Children** **(1 year to 12 years)**	1260 (76.88%)	1379 (72.46%)	1973 (74.23%)	2026 (68.33%)	2018 (66.84%)	1996 (62.1%)	571 (37.15%)	620 (54.01%)	538 (54.12%)	656 (62.12%)
	**Adolescents** **(13 years to 17 years)**	24 (1.46%)	24 (1.26%)	34 (1.28%)	40 (1.35%)	43 (1.42%)	54 (1.68%)	41 (2.67%)	17 (1.48%)	21 (2.11%)	20 (1.89%)
	**Adults** **(18 years to 64 years)**	75 (4.58%)	112 (5.89%)	235 (8.84%)	354 (11.94%)	358 (11.86%)	554 (17.24%)	394 (25.63%)	178 (15.51%)	171 (17.20%)	168 (15.91%)
	**Older adults** **(65 years or older)**	69 (4.21%)	121 (6.36%)	156 (5.87%)	227 (7.66%)	267 (8.84%)	383 (11.92%)	395 (25.70%)	211 (18.38%)	162 (16.30%)	145 (13.73%)
**Gender**	**Male**	887 (54.12%)	1025 (53.86%)	1407 (52.93%)	1539 (51.91%)	1600 (53.00%)	1650 (51.34%)	793 (51.59%)	612 (53.31%)	502 (50.50%)	564 (53.41%)
	**Female**	752 (45.88%)	878 (46.14%)	1251 (47.07%)	1426 (48.09%)	1419 (47.00%)	1564 (48.66%)	744 (48.41%)	536 (46.69%)	492 (49.50%)	492 (46.59%)
**Patient type**	**Inpatients**	1327 (80.96%)	1454 (76.41%)	1917 (72.12%)	2141 (72.21%)	2077 (68.80%)	2007 (62.45%)	866 (56.34%)	746 (64.98%)	445 (44.77%)	471 (44.60%)
	**Outpatients**	312 (19.04%)	449 (23.59%)	741 (27.88%)	824 (27.79%)	942 (31.20%)	1207 (37.55%)	671 (43.66%)	402 (35.02%)	549 (55.23%)	585 (55.40%)

* Data collected from 18 January 2014 to 30 April 2023. # Age ranges with reference to US National Institutes of Health (NIH) [13].

**Table 2 viruses-15-01820-t002:** Co-detection of respiratory viruses from January 2014 to April 2023.

	CoVs	RV/EV	MPV	Flu A & B	PIV1–4	RSV	SARS-CoV-2
**ADV**	71	**413**	39	53	142	72	2
**CoVs**		186	20	78	90	50	0
**RV/EV**			181	335	**672**	**396**	6
**MPV**				35	62	28	1
**Flu A & B**					93	58	4
**PIV1–4**						147	3
**RSV**							2

**Table 3 viruses-15-01820-t003:** Per-virus PCR positivity in lower respiratory tract (LRT), upper respiratory tract (URT), and combined URT specimens.

	LRT				URT				Combined URT			
	Detected		Not Detected		Detected		Not Detected		Detected		Not Detected	
	*n*	%	*n*	%	*n*	%	*n*	%	*n*	%	*n*	%
**Flu A & B**	35	5.93	555	94.07	2763	**16.62**	13,861	83.38	69	2.37	2844	97.63
**PIV1–4**	12	2.03	578	97.97	1769	**10.64**	14,855	89.36	192	6.59	2721	93.41
**RSV**	12	2.03	578	97.97	1573	**9.46**	15,051	90.54	231	7.93	2682	92.07
**MPV**	14	2.37	576	97.63	882	**5.31**	15,742	94.69	30	1.03	2883	98.97
**ADV**	15	2.54	575	97.46	1337	**8.04**	15,287	91.96	48	1.65	2865	98.35
**RV/EV**	73	12.37	517	87.63	4594	**27.63**	12,030	72.37	561	19.26	2352	80.74
**CoVs**	12	2.03	578	97.97	724	**4.36**	15,900	95.64	69	2.37	2844	97.63
**SARS-CoV-2**	10	**8**	115	92	20	2.53	771	97.47	127	5.77	2074	94.23

**Remarks:** (1) Total number of specimens: 20,127; LRT: 590; URT: 16,624; combined URT: 2913; (2) Number of specimens with SARS-CoV-2 data: 3117; LRT: 125; URT: 791; combined URT: 2201.

## Data Availability

The data presented in this study are available on request from the corresponding author.

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
