# Peer review of "The Seasonality of Respiratory Viruses in a Hong Kong Hospital, 2014–2023"

_viruses, 2023, doi:10.3390/v15091820_

Round 1
Reviewer 1 Report
In the manuscript by Wai Sing Chan et al, the authors describe results from single center retrospective/observational study, evaluating seasonality of respiratory viruses from 2014 to 2023.
Nevertheless, the argument is interesting, but the manuscript needs to be increased of data and details.
Materials and methods: were the samples “unique samples” or “duplicate sample”? what were the criteria of inclusion and/or exclusion? After how many times a sample of a patient could be re-include in the study? I would clarify that to avoid any bias for the proportion of respiratory viruses.
Figure 1: “In addition, we also considered the data of Middle East respiratory syndrome coronavirus (MERS-CoV) and severe acute respiratory syndrome coronavirus 2 (SARS-75 CoV-2) from January 2021 to April 2023 which are additional targets in newer versions of FA RP”. Please double check in the left part of figure 1, because the gray bars of SARS-CoV-2 appear only in 2022? Is correct?
Results: In the paragraph “background demographics” and in table 1 please evaluate a non-overlapping age range maybe should be infant (2 months to 1 year) and children (2 years to 12 years)?
In paragraph “trends in the number of testing and PCR positivity” I would add a minimal statistical analysis between the different averages; would give more robustness to the manuscript.
The English language is clear and essential. I would perfection only some passage to improve the readability of this manuscript.
Author Response
Reviewer 1
Comment 1:
Materials and methods: were the samples “unique samples” or “duplicate sample”? what were the criteria of inclusion and/ or exclusion? After how many times a sample of a patient could be re-include in the study? I would clarify that to avoid any bias for the proportion of respiratory viruses.
Response to Comment 1:
Thank you for the very good comment. The inclusion/ exclusion criteria were (1) multiplex PCR data, (2) respiratory specimens, (3) specimens collected from 18 January 2014 to 30 April 2023 and tested in our hospital and (4) specimens with indeterminate result were excluded. Points 1-3 are described in line 47-54 and 69-70. For point 4, we have added ‘Specimens with indeterminate result were excluded.’ in line 70 for clarity.
As the personal identifiers were removed prior to data analysis (line 70-71), we included all the specimens fulfilling the above selection criteria for data analysis. Considering the duration of PCR positivity varies with different virus types, it is challenging to set a common temporal cutoff for optimal inclusion/ exclusion of specimens from the same patient. We have stated this limitation in line 253-255: ‘Another source of sampling bias might originate from ‘duplicate’ specimens of the same patient collected during the same phase of PCR positivity’.
*************************************************************************
Comment 2:
Figure 1: “In addition, we also considered the data of Middle East respiratory syndrome coronavirus (MERS-CoV) and severe acute respiratory syndrome coronavirus 2 (SARS-75 CoV-2) from January 2021 to April 2023 which are additional targets in newer versions of FA RP”. Please double check in the left part of figure 1, because the gray bars of SARS-CoV-2 appear only in 2022? Is correct?
Response to Comment 2:
We checked the multiplex PCR data again and confirmed the very low PCR positivity of SARS-CoV-2 in 2021. We attempted to account for this phenomenon in line 251-253: ‘To a certain extent, the drastic shift in the demand towards SARS-CoV-2-specific PCR might introduce sampling bias, which might explain the very low prevalence of SARS-CoV-2 in 2021’.
*************************************************************************
Comment 3:
Results: In the paragraph “background demographics” and in table 1 please evaluate a non-overlapping age range maybe should be infant (2 months to 1 year) and children (2 years to 12 years)?
Response to Comment 3:
We have modified the age ranges of infants (1 month to <1 year) and neonates or newborns (birth to <1 month) to better deliver the meaning (line 110 and Table 1).
*************************************************************************
Comment 4:
In paragraph “trends in the number of testing and PCR positivity” I would add a minimal statistical analysis between the different averages; would give more robustness to the manuscript.
Response to Comment 4:
We have compared the difference between the observed means in the number of testing and PCR positivity. These parameters were significantly lower during the pandemic (2020-2022) compared with pre-COVID-19 period (2014-2019) and 2023 (p < 0.001) (line 127-129).

Reviewer 2 Report
In their brief report, Wai Sing Chan et al. show the result of a thorough analysis of multiplex PCR assays of respiratory samples in the period 2014-2023, consequently including the COVID-19 pandemic period.
The article is well written and flows smoothly.
I have some minor comments.
Introduction
- Line 17, outbreaks (plural rather than singular, as you are referring to two outbreaks)
- Line 24, replace “was” with “is”
Figure 1: I would place it after the methods.
Results
- Line 90: I would eliminate “The USA”, and write instead “… with reference to US National Institute of Health (NIH)”
- Paragraph 3.5.7.: I would remind to the reader that the reason why the detection frequency of coronavirus, included SARS-CoV-2, is close to zero during the first two year of pandemic, is that you are referring to a multiplex PCR (otherwise it seems an impossible result)
Discussion
I would move sentences from lines 203 to 209 and from 229 to 231 at the end of the paper as they represent the conclusion.
Author Response
Reviewer 2
Comment 1:
Line 17, outbreaks (plural rather than singular, as you are referring to two outbreaks)
Response to Comment 1:
We have made the change accordingly (line 25).
*************************************************************************
Comment 2:
Line 24, replace “was” with “is”
Response to Comment 2:
We have made the change accordingly (line 32).
*************************************************************************
Comment 3:
Figure 1: I would place it after the methods.
Response to Comment 3:
We have made the change accordingly (P.3).
*************************************************************************
Comment 4:
Line 90: I would eliminate “The USA”, and write instead “… with reference to US National Institute of Health (NIH)”
Response to Comment 4:
We have made the change accordingly (line 108-109 & 116).
************************************************************************* Comment 5:
Paragraph 3.5.7.: I would remind to the reader that the reason why the detection frequency of coronavirus, included SARS-CoV-2, is close to zero during the first two year of pandemic, is that you are referring to a multiplex PCR (otherwise it seems an impossible result)
Response to Comment 5:
We have made the change accordingly: ‘Detection frequency of CoVs by multiplex PCR, including SARS-CoV-2, was very low…’ (line 198-199).
*************************************************************************
Comment 6:
I would move sentences from lines 203 to 209 and from 229 to 231 at the end of the paper as they represent the conclusion.
Response to Comment 6:
Thank you for the very good suggestion. Considering the flow of the manuscript, we have added the conclusion section (line 262-267) instead and hope you would accept the changes.

Reviewer 3 Report
This study presents interesting data on the seasonal oscillation patterns of respiratory viruses in Hong Kong. The main research findings of this paper will be important for public health and infection control. I have just a few comments, mainly on writing style, which should help to improve the clarity of the contribution.
Specific comments:
Abstract: Your data showed the frequency of detection with FluA, FluB, PIV, RSV, ADV and CoVs (expected SARS-CoV-2) were decreased during first 2 years of COVID-19 pandemic and but the frequency of detection with RV/EV was quite a few. In addition, the suppressed viruses appeared to regain activity from Q4 of 2022. I think you need to describe the above into Abstract.
Line 50-54: It is different order between Figure legend and graph legend. It is better to change the order for easier understanding.
Line 56: The authors would say whether ethical review board approved.
Line 194-195: PCR positivity and the number of specimens decreased during the first 3 years of pandemic. If possible, the authors need to explain the social background and reasons.
For example:
-There was a situation where people didn’t want to go to hospital because overwhelmed health care system.
-There is to refrain from seeing a doctor.
If there was happened the above, the authors would write in limitation section.
I hope these comments will be helpful.
Author Response
Reviewer 3
Comment 1:
Abstract: Your data showed the frequency of detection with FluA, FluB, PIV, RSV, ADV and CoVs (expected SARS-CoV-2) were decreased during first 2 years of COVID-19 pandemic and but the frequency of detection with RV/EV was quite a few. In addition, the suppressed viruses appeared to regain activity from Q4 of 2022. I think you need to describe the above into Abstract.
Response to Comment 1:
We have modified the abstract accordingly (line 11-19).
*************************************************************************
Comment 2:
Line 50-54: It is different order between Figure legend and graph legend. It is better to change the order for easier understanding.
Response to Comment 2:
We have made the changes accordingly (line 97-99).
*************************************************************************
Comment 3:
Line 56: The authors would say whether ethical review board approved.
Response to Comment 3:
This study was approved by Research Ethics Committee of our hospital (line 274-275).
*************************************************************************
Comment 4:
Line 194-195: PCR positivity and the number of specimens decreased during the first 3 years of pandemic. If possible, the authors need to explain the social background and reasons.
For example:
-There was a situation where people didn’t want to go to hospital because overwhelmed health care system.
-There is to refrain from seeing a doctor.
If there was happened the above, the authors would write in limitation section.
Response to Comment 4:
We have added the sentence ‘On the other hand, as current study was based on observation of laboratory data, we could not determine the exact social factors contributing to the decreased number of specimens and PCR positivity during the pandemic.’ in line 255-258 to describe this limitation.
